# Prevalence of Viral Frequency-Dependent Infection in Coastal Marine Prokaryotes Revealed Using Monthly Time Series Virome Analysis

Kento Tominaga,[a,b] Nana Ogawa-Haruki,[a] Yosuke Nishimura,[c] Hiroyasu Watai,[a] Keigo Yamamoto,[d] Hiroyuki Ogata,[e] Takashi Yoshida[a]

aGraduate School of Agriculture, Kyoto University, Kyoto, Japan
bGraduate School of Frontier Sciences, The University of Tokyo, Tokyo, Japan
cJapan Agency for Marine-Earth Science and Technology (JAMSTEC), Kanagawa, Japan
dResearch Institute of Environment, Agriculture and Fisheries, Osaka Prefecture, Osaka, Japan
eInstitute for Chemical Research, Kyoto University, Kyoto, Japan

**ABSTRACT** Viruses infecting marine prokaryotes have a large impact on the diversity and dynamics of their hosts. Model systems suggest that viral infection is frequency dependent and constrained by the virus-host encounter rate. However, it is unclear whether frequency-dependent infection is pervasive among the abundant prokaryotic populations with different temporal dynamics. To address this question, we performed a comparison of prokaryotic and viral communities using 16S rRNA amplicon and virome sequencing based on samples collected monthly for 2 years at a Japanese coastal site, Osaka Bay. Concurrent seasonal shifts observed in prokaryotic and viral community dynamics indicated that the abundance of viruses correlated with that of their predicted host phyla (or classes). Cooccurrence network analysis between abundant prokaryotes and viruses revealed 6,423 cooccurring pairs, suggesting a tight coupling of host and viral abundances and their "one-to-many" correspondence. Although stable dominant species, such as SAR11, showed few cooccurring viruses, a fast succession of their viruses suggests that viruses infecting these populations changed continuously. Our results suggest that frequency-dependent viral infection prevails in coastal marine prokaryotes regardless of host taxa and temporal dynamics.

**IMPORTANCE** There is little room for doubt that viral infection is prevalent among abundant marine prokaryotes regardless of their taxa or growth strategy. However, comprehensive evaluations of viral infections in natural prokaryotic communities are still technically difficult. In this study, we examined viral infection in abundant prokaryotes by monitoring the monthly dynamics of prokaryotic and viral communities at a eutrophic coastal site, Osaka Bay. We compared the community dynamics of viruses with those of their putative hosts based on genome-based *in silico* host prediction. We observed frequent cooccurrence among the predicted virus-host pairs, suggesting that viral infection is prevalent in abundant prokaryotes regardless of their taxa or temporal dynamics. This likely indicates that frequent lysis of the abundant prokaryotes via viral infection has a considerable contribution to the biogeochemical cycling and maintenance of prokaryotic community diversity.

**KEYWORDS** frequency-dependent selection, *K/r* strategy, marine prokaryotes, marine viruses, virome

Address correspondence to Takashi Yoshida, yoshida.takashi.7a@kyoto-u.ac.jp, or Hiroyuki Ogata, ogata@kuicr.kyoto-u.ac.jp.

The authors declare no conflict of interest.

Marine prokaryotes are ubiquitous in the ocean and play key roles in global biogeochemical processes (1). Most of the observed species (>35,000 species-level operational taxonomic units [OTUs] based on 97% 16S rRNA sequence identity) fall into several

major taxa (phyla or classes for *Proteobacteria*), such as *Alphaproteobacteria* (e.g., SAR11), *Bacteroidetes* (e.g., *Flavobacteriaceae*), and *Cyanobacteria* (e.g., *Synechococcus* and *Prochlorococcus*) (2, 3). Although individual species have distinct ecological niches, they are often classified into one of two growth strategists based on their potential growth rate and temporal dynamics: (i) *K*-strategists (slow growing and persistently dominant; e.g., SAR11) and (ii) *r*-strategists (fast growing and opportunistic; e.g., *Flavobacteriaceae*) (4). However, recent high-frequency sampling schemes (e.g., daily) uncovered that species not recognized as *r*-strategists exhibit drastic fluctuations (e.g., Marine Group II [MGII] *Euryarchaeota*) (5, 6). Further, finely resolved populations (genotypes or strains) within a species-level OTU often show distinct temporal dynamics (7–11), indicating that species described as *K*-strategists can show frequent fluctuation.

Viruses infecting prokaryotes are abundantly present in the ocean and are estimated to lyse 20 to 40% of the prokaryotic cells each day (4, 12, 13). Viruses are thought to infect their specific hosts (often restricted to strains within a species) in a frequency-dependent manner, in which the encounter rate between the viruses and their hosts is a determinant for the infection rate (14, 15). Thus, viruses infect host populations that become abundant, and frequencies of host and viruses oscillate over time, leading to maintenance of the diversity of the host community (16, 17). Moreover, mathematical models have demonstrated that a prokaryotic species with a higher growth rate can be more susceptible to viral infection (17). This trend allows *K*-strategists to reach a higher abundance than *r*-strategists because of their higher resistance against viral infection by cryptic escape through reduced cell size and/or specialized defense mechanisms (4, 18). However, the discovery of SAR11 viruses questions this prediction (19). Given the dominance of SAR11 viruses in the ocean (19–21), there is little room for doubt that *K*-strategists are also targeted by viruses. Therefore, comprehensive surveys to examine the prevalence of viral infections in abundant prokaryotes with different growth strategies are still required.

Previous monthly observations of microbial communities revealed that seasonal oceanographic features have a strong influence on the prokaryotic community (22, 23). Seasonal variability of the viral community has also been reported using PCR-based analysis (24, 25) and viral metagenomics (viromics) (26–28). Although viruses are obligate parasites, viral seasonality was often discussed independently from the seasonality of their hosts except for a few well-cultivated prokaryote-virus pairs (e.g., *Synechococcus*/*Prochlorococcus* and their viruses) (25, 29). Otherwise, the interactions are described solely based on the dynamics of individual viruses and prokaryotes because of the difficulty in connecting uncultured viruses and their hosts (13, 30).

In this study, we aimed to solve the fundamental question of whether viral infection is prevalent among abundant prokaryotic populations or the way viruses infect differs depending on the taxa and/or growth strategies of their hosts. For this purpose, we monitored prokaryotic and viral communities at Osaka Bay, a eutrophic coastal site with input of nutrients from rivers that is affected by an oligotrophic warm current Kuroshio (31), for 2 years at a monthly time interval. In the sampling site, monitoring of microbial communities (32) and characterization of full-length double-stranded DNA (dsDNA) viral genomes assembled from virome sequences (33) and their diurnal transcriptomic dynamics (34, 35) have been reported. However, cooccurrence dynamics alone have been suggested to be a poor predictor of interactions in the context of viral infection of microbial hosts (36).To overcome the limitations of correlation-based inference of virus-host pairs, we compare the community dynamics of viruses with those of their putative hosts using genome-based *in silico* host prediction analysis (29, 30). The prevalence of viral infection is discussed with cooccurrence dynamics among the potential virus-host pairs, which fulfilled the genome-based prediction.

## RESULTS AND DISCUSSION

**Overview of prokaryotic and viral communities in Osaka Bay.** We collected more than 1 year of time series (total of 18 months) seawater samples during the daytime (3 h before or after high tide) of each month. We obtained 2.8 million paired-end

reads (24,168 to 846,565 reads per sample) from the 16S rRNA gene V3-V4 region amplicon sequencing libraries, and these sequences were clustered into 35,191 OTUs (1,462 to 18,268 OTUs per month, median of 3,274) with a sequence identity threshold of 99% (species-level populations are presented in Table S1 in the supplemental material). The prokaryotic community was dominated by *Alphaproteobacteria* (41%), *Gammaproteobacteria* (21%), *Bacteroidetes* (19%), and *Cyanobacteria* (7%) at the phylum level (class level for *Proteobacteria*).

To explore viral community composition, we obtained 60 million paired-end reads of viromes (929,884 to 8,124,354 sequences per sample), which were generated from the virus-size fraction of 17 samples that were concomitantly collected with the prokaryotic-size fractions (Table S1). After decontamination of prokaryotic sequences and dereplication, 5,226 virus-like large contigs (>10 kb), including 202 circularly assembled viral genomes, were obtained (Table S1). Here, we call the 5,226 contigs assembled in this study monthly time series Osaka Bay viral (mts-OBV) contigs. We refer to these contigs (or genomes) operationally as species-level viral populations, according to the previous proposal in viral ecology (37). The majority (~75%) of mts-OBV contigs showed high genomic similarity (genomic similarity score ($S_G$) of >0.15; see reference 38 for the definition of $S_G$) with one of the previously reported viral complete genomes (33) or the 202 circular genomes assembled in this study. Based on the $S_{G,}$ these mts-OBV contigs were classified into 314 genus-level taxonomic groups (Table S1).

On average, 40% of virome reads (29 to 53% per sample) were mapped on the mts-OBV contigs and previously reported viral genomes (33). According to the fragments per kilobase per million (FPKM) values calculated from the read counts, mts-OBV contigs (assembled in this study) occupied, on average, 96% of the relative abundance of analyzed viral genomes, suggesting mts-OBVs well represent the viral community observed in the samples. Other relatively abundant genomes included genomes assembled in a previous study at Osaka Bay (e.g., OBV_N00129 and OBV_N00081) (33) and *Pelagibacter* virus HTVC010P (19). Further, we confirmed whether the mts-OBV contigs can cover the trend of the whole viral community by the following procedure according to the previous study (34). The abundances of mts-OBV and other smaller contigs (1 to 10 kb) assembled in this study were compared based on the abundances of all detected terminase large subunit genes (*terL*). mts-OBV contigs covered a wide range of abundance, suggesting that they can capture the trend of the whole viral community. In addition, all mts-OBV contigs ranked at the top (>30%) of the community in at least one sample (Fig. S1), indicating that most abundant viruses were captured within mts-OBVs. Thus, in the following sections, only the 5,226 mts-OBV contigs (including 202 complete genomes) and 4,240 previously reported complete viral genomes (33) were considered viral community assessment for read mapping.

Alpha-diversity (Shannon index) of the viral community was higher than that of the prokaryotic community (Fig. S2A and B). Both richness and evenness were also higher in the viral community than in the prokaryotic community (Fig. S2C to F). It should be noted that prokaryotic diversity was evaluated via single marker gene analysis (i.e., 16S rRNA), but viral diversity was evaluated via whole-genome sequencing. Thus, the methodological difference could have caused the relatively higher diversity of the viral community, such as double counting of partial contigs derived from a viral genome. Another possible explanation for the higher viral diversity is that a prokaryotic species can be infected by more than one viral species at each time point (discussed below).

**Seasonal dynamics of prokaryotic and viral communities.** We investigated seasonal dynamics of prokaryotic and viral communities using the Bray-Curtis similarity index between all pairwise combinations of samples (136 pairs, 1- to 17-month intervals), following the previous time series observations in different locations (22, 25, 26, 39, 40). Both prokaryotic and viral communities showed clear seasonal patterns, with a peak average similarity at an interval of about 12 months, representing the same seasons, and the bottom of average similarity at an interval of 6 months, representing opposite seasons (Fig. 1). Similar seasonal peaks have been observed in previous studies,

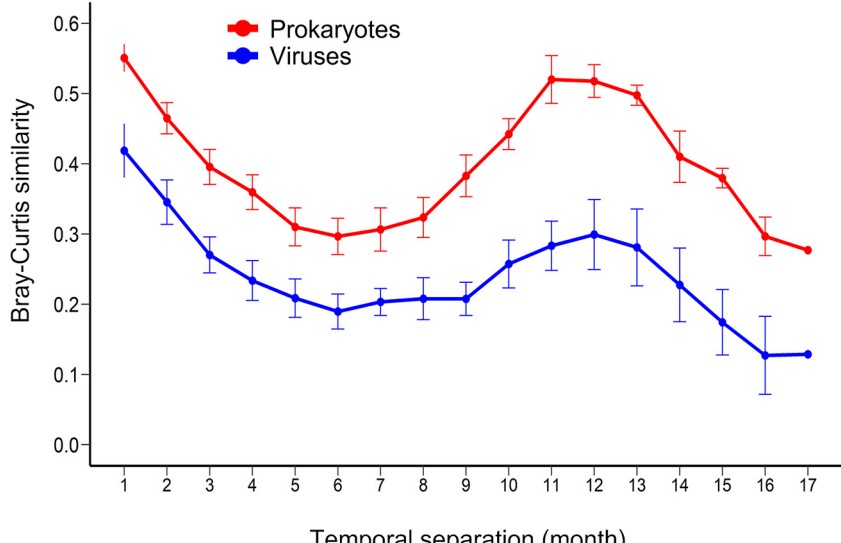

**FIG 1** Seasonality of the prokaryotes and viruses at Osaka Bay (OB) during observation. The Bray-Curtis community similarity index was calculated among all of the possible sample pairs from normalized abundances of prokaryotic OTUs and OBV contigs and was plotted as a function of the number of months separating their sampling.

such as the San-Pedro Ocean Time (SPOT) series conducted in San Pedro Channel (22, 25, 26, 39–41). However, the lower community similarity observed between opposite seasons in both prokaryotic and viral communities at Osaka Bay than observed at SPOT using a similar methodology (26, 41) suggests more pronounced seasonality at Osaka Bay. Prokaryotic community dynamics were concordant with seasonal environmental variables, such as water temperature and inorganic nutrients, which increased in summer (June to September) presumably because of the increasing river inflow during the rainy season (Table 1; Table S1). The community variation is generally larger (low similarity) between the autumn and spring samples than between the summer and winter samples in both communities (Fig. S3). The similarity between samples was systematically lower for the viral community than for the prokaryotic community (Fig. 1, discussed below). The viral community composition was significantly correlated with the prokaryotic community composition as well as the seasonal environmental variables (Mantel, $\rho = 0.504$, $P < 0.01$; Table 1).

Given that most viruses can only propagate in their specific host and thereby the viral community composition is shaped by the prokaryotic community composition, the

**TABLE 1** Rho values of partial Mantel tests for prokaryotic and viral communities and environmental parameters

| | Prokaryotes[a] | Viruses | Environmental combined | Nutrient combined | Temp | NH$_4$ | NO$_2$ | NO$_3$ | PO$_4$ | Salinity |
|---|---|---|---|---|---|---|---|---|---|---|
| Prokaryotes | | | | | | | | | | |
| Viruses | 0.504[c] | | | | | | | | | |
| Environmental combined | 0.308[c] | 0.227[b] | | | | | | | | |
| Nutrient combined | 0.344[c] | 0.288[c] | 0.871[c] | | | | | | | |
| Temp | 0.505[c] | 0.137 | 0.37[c] | 0.284[c] | | | | | | |
| NH$_4$ | −0.097 | −0.109 | 0.328[c] | 0.458[c] | 0.01 | | | | | |
| NO$_2$ | 0.037 | −0.015 | 0.379[c] | 0.53[c] | 0.007 | 0.404[c] | | | | |
| NO$_3$ | 0.239[b] | 0.3[c] | 0.206[b] | 0.34[c] | −0.044 | −0.113 | −0.06 | | | |
| PO$_4$ | 0.511[c] | 0.443[c] | 0.487[c] | 0.56[c] | 0.154 | 0.035 | 0.108 | 0.218[b] | | |
| Salinity | −0.159 | −0.037 | 0.502[c] | 0.191 | −0.028 | 0.078 | 0.01 | −0.16 | 0.115 | |
| SiO$_2$ | 0.138 | 0.078 | 0.565[c] | 0.329[c] | 0.447[c] | 0.028 | −0.037 | −0.063 | 0.042 | 0.459[c] |

[a]The value in each box is the Rho value.
[b]$P < 0.05$.
[c]$P < 0.01$.

abundance of each virus might reflect the abundance of its host. To test this hypothesis, compositions of prokaryotic and viral communities were compared using the information of predicted viral hosts (mostly host phylum- or class-level composition). Putative host groups of viruses were predicted using four commonly used genome-based *in silico* prediction methods (similarity with known viruses, CRISPR-spacer match, tRNA match, and genome homology). First, based on the similarity with cultured viruses, putative host groups of 951 mts-OBV contigs (22 genomic OTUs [gOTUs]) were predicted (*Synechococcus/Prochlorococcus*, 182 contigs; SAR11, 501 contigs; SAR116, 214 contigs; *Roseobacter*, 31 contigs; others, 23 contigs; Table S1). Similarly, putative host groups of 504 mts-OBV contigs (39 genera) were predicted based on the genome-wide sequence similarities with uncultured viral genomes according to the previous studies (*Bacteroidetes*, 468 contigs; MGII, 36 contigs [33, 42]; Table S1). For the other 1,460 mts-OBV contigs (*Alphaproteobacteria*, 35 gOTUs, 621 contigs; *Bacteroidetes*, 80 contigs; *Gammaproteobacteria*, 236 contigs; *Deltaproteobacteria*, 326 contigs; others, 53 contigs; Table S1), putative host groups were predicted via the sequence similarity (i.e., CRISPR-spacer matching, tRNA matching, and genome homology) between viral (mts-OBVs with >200,000 previously reported marine viral genomes [26, 33, 38, 43, 44]) and prokaryotic genomic data sets (>8,000 marine prokaryotic metagenome-assembled genomes in previous studies [45–49] and the genomes in the NCBI RefSeq database). Altogether, we assigned potential host groups for 2,844 mts-OBV contigs (*Alphaproteobacteria*, 1,375 contigs; *Bacteroidetes*, 548 contigs; *Deltaproteobacteria*, 326 contigs; *Gammaproteobacteria*, 250 contigs; *Cyanobacteria*, 190 contigs; Table S1).

Major phyla (or classes for *Proteobacteria*) in the prokaryotic community did not change drastically, but the relative abundance of several phyla (classes) exhibited seasonal dynamics (Fig. 2). The seasonal dynamics of the predicted hosts resembled the seasonal dynamics of prokaryotes (Fig. 2). For example, *Cyanobacteria* (79% of reads were assigned to OTU_8, *Synechococcus*) dominated in summer (up to 9.6% and 22.6% of the community in June 2015 and July 2016, respectively) (Fig. 2), and *Synechococcus* virus abundance also increased in summer (up to 5.3% and 12.1% of the community in August 2015 and August 2016, respectively) (Fig. 2). Similarly, the relative abundance of *Bacteroidetes* increased from winter to spring (up to 33.7% of the community in May 2016) (Fig. 2), and *Bacteroidetes* virus abundance also increased during spring (up to 30.2% of the community in May 2016) (Fig. 2). Relative abundances of both SAR11 (from 5% to 47% of the community) (Fig. 2) and SAR11 viruses (from 9% to 22% of the community) (Fig. 2) showed changes over time, but they were always abundant throughout the observed period. Therefore, the viral community appears to generally follow the dynamics of their host.

However, viral abundance did not always match with their putative host abundance (Fig. S4). For example, the proportion of putative *Gammaproteobacteria* viruses was lower than that of *Gammaproteobacteria*, and the proportion of putative *Deltaproteobacteria* viruses was much higher than that of *Deltaproteobacteria* (Fig. 2). The lack of a tight correlation between viral and host abundance may not be surprising. Host prediction based on genome analysis in this study was mostly at the phylum or class level except for contigs showing similarity with cultured viruses, such as *Synechococcus/Prochlorococcus* cyanoviruses, while typical prokaryotic viruses could only infect specific host species or strains. Further, although our analysis annotated putative hosts at nearly 60% of the viral community, remaining populations without host prediction may lead to the underestimation of viruses infecting some taxa. The differences in burst sizes among viruses, which have been estimated to range from 6 to 300 in the marine environment (50), can also influence the estimation of viral abundance. Alternatively, as previously observed and discussed in Hevroni et al. (28), differences in viral infection cycles (e.g., consistently low production of virions [51]) or short time delay of peaks of virus and host abundance (52) might cause the mismatch between host abundance and extracellular viral abundance. Next, to investigate whether viral abundance increased according to specific host abundance, we statistically examined associations (i.e., cooccurrence) between the viruses and amplicon sequence variants (ASVs) extracted from the abundant 73 prokaryotic OTUs.

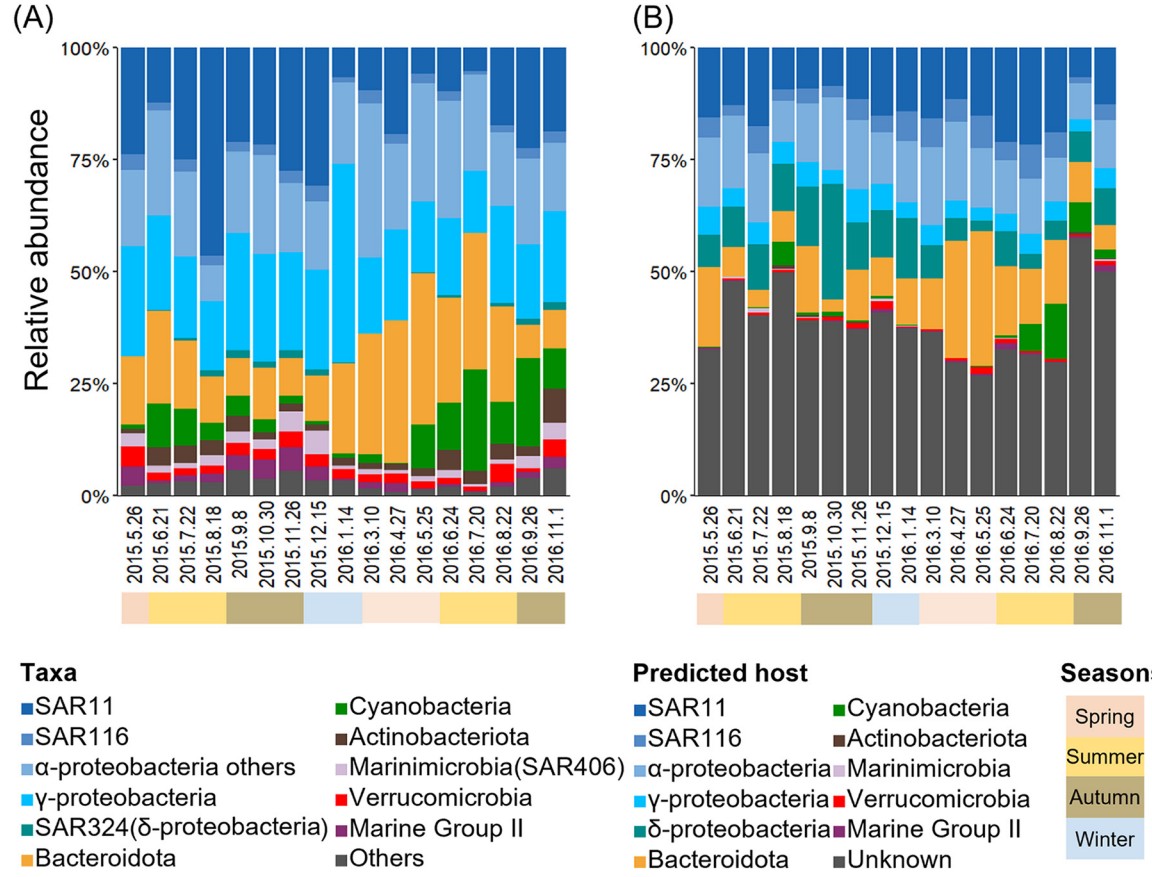

**FIG 2** Comparison of prokaryotic and viral taxonomic community composition based on host prediction. (A) Relative abundance of phylogenetic groups of prokaryotic communities. Quality-controlled reads were clustered into OTUs with a sequence identity of 99% using VSEARCH (73). These OTUs were classified at the phylum level (class level for *Proteobacteria*) using SINA (75). (B) Relative abundance of viruses based on their putative hosts assigned by host prediction. Normalized abundances of viral contigs were calculated from fragments per kilobase per million (FPKM) values and were converted to relative abundance.

**Cooccurrence network analysis between the abundant prokaryotes and viruses.** To examine the dynamics of closely related (nearly strain-level) variants within each OTU, 114 ASVs (1~4 ASVs per OTU) (Fig. S5) were extracted from the abundant 74 OTUs via minimum entropy decomposition (MED), according to previous studies (7, 10, 11). Although the majority of ASVs derived from the same abundant OTU showed similar dynamics, several ASVs showed distinct seasonal patterns (e.g., ASVs from SAR86 OTU2 and NS4 marine group *Flavobacteria* OTU14) (Fig. S5). These ASVs likely reflect ecologically meaningful seasonal subpopulations in an OTU that dominates throughput the sampling period. Then, pairwise correlations (cooccurrence network) between the 114 prokaryotic ASVs and the viral species that were predicted to infect the prokaryotic ASVs via host prediction (e.g., 37 *Bacteroidetes* ASVs and 548 mts-OBV contigs predicted as *Bacteroidetes* virus) were determined via Spearman's correlations. In total, 6,423 significant correlations between 104 prokaryotic ASVs and 1,366 viral species were detected (Fig. 3; Fig. S6A). The majority (88.6%) of prokaryotic ASVs correlated with at least one viral species. In contrast, only 34% and 31% of prokaryotic ASVs positively and negatively correlated with environmental variables, respectively (Spearman correlations, $r > 0.6$, $P < 0.01$, $q < 0.05$) (Table S2). The number of cooccurring viral species ranged from 0 (13 ASVs) to 359 (ASV6-1, classified into *Planktomarina*), and the median value was 16. We verified that the correlation is not only because the pattern of viral and host dynamics was similar but also because the virus ranked high in the viral community when the putative host was abundant. We therefore compared the relative rank among the viral community and the host prokaryotic abundance in the detected 6,423

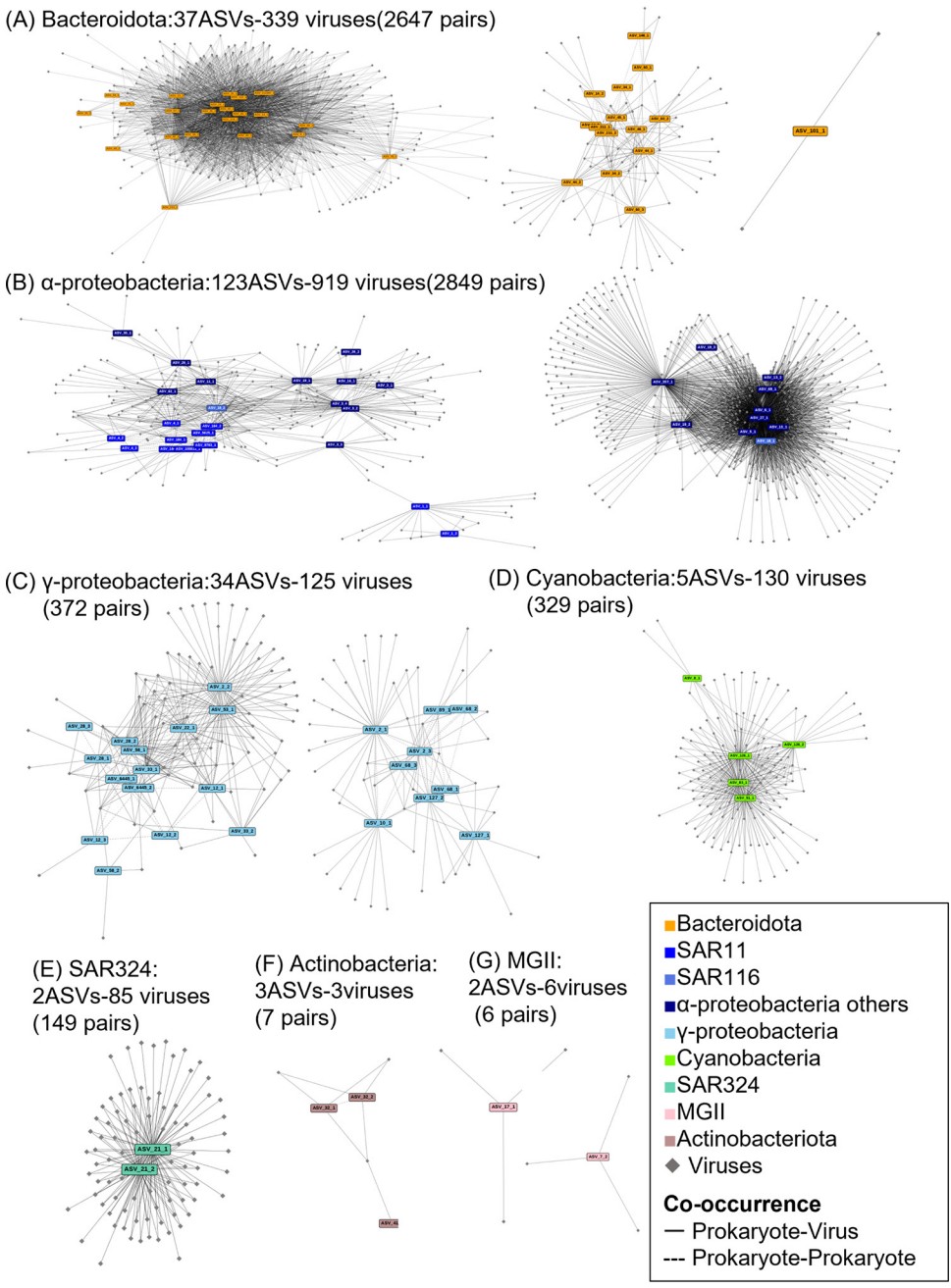

**FIG 3** A broad overview of detected positive correlations between prokaryotic ASVs and viral populations that potentially infect each prokaryotic taxa based on host prediction analysis. (A) *Bacteroidota* and their viruses. (B) *Alphaproteobacteria* and their viruses. (C) *Gammaproteobacteria* and their viruses. (D) *Cyanobacteria* and their viruses. (E to G) Other major groups, such as SAR324 (E), *Actinobacteria* (F), and Marine Group II (G), and their viruses. Prokaryotic nodes are circles, and viral nodes are V shapes. Node color indicates prokaryotic taxa. Solid lines are positive correlations.

putative virus-host pairs. First, four cyanobacterial ASVs and cooccurring 130 cyanovirus species were examined. Because substantial numbers of *Synechococcus/Prochlorococcus*-virus pairs have been reported in culture-based studies (53–56), host prediction for cyanoviruses is likely to be reliable. These cyanoviral species were more dominant in the viral community when their cooccurring ASVs exceeded the predicted minimum host cell density for effective propagation of prokaryotic viruses ($10^3$ cells/mL [57] or $10^4$ cells/mL [58]) (Fig. 4; Fig. S6B and C). Thus, cyanobacterial viral species were not abundant or were often undetectable when their putative hosts were less abundant, but they became dominant

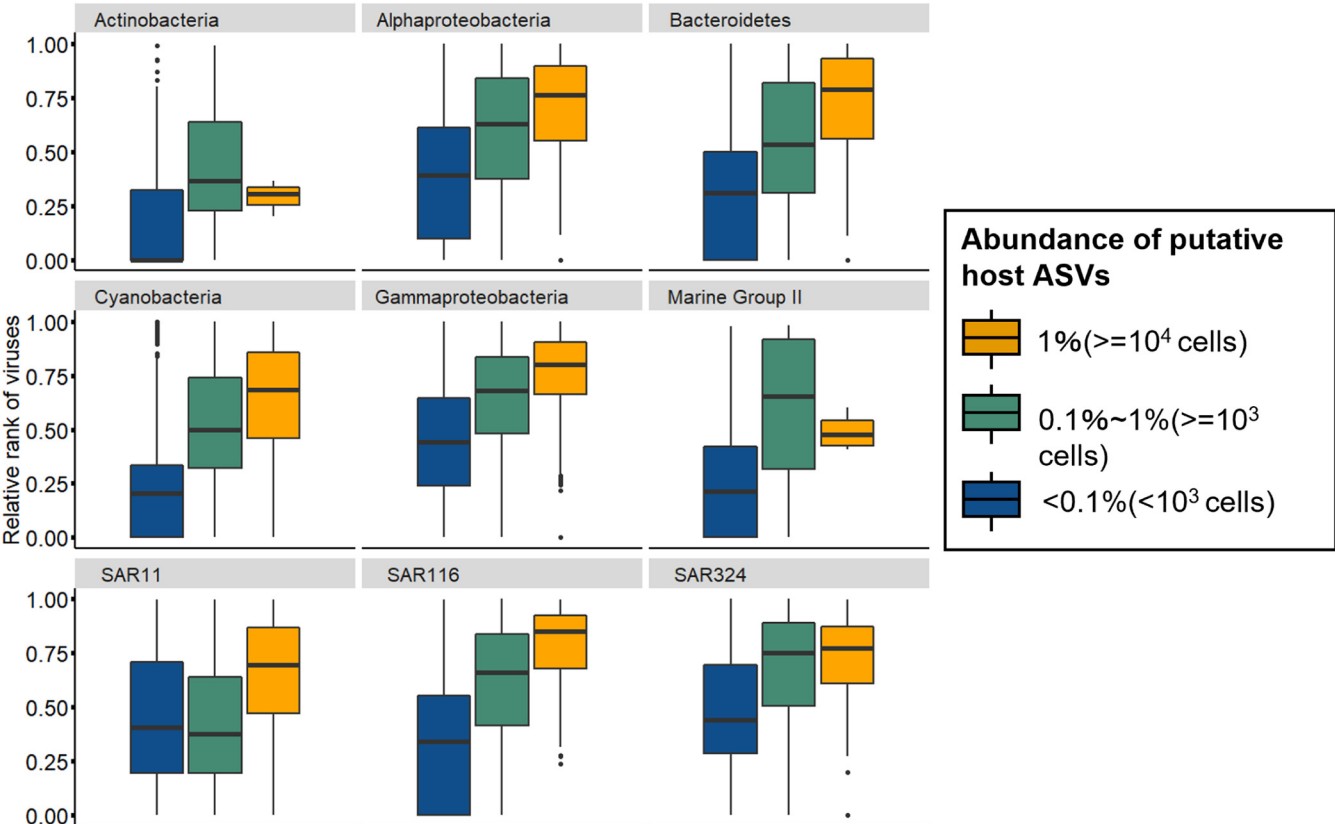

**FIG 4** Increase of viral abundance according to host cell density between cooccurring host-virus pairs. The normalized relative rank of each virus in the community (0 to 1) was plotted when their putative host relative abundance exceeded 1% ($\approx 10^4$ cells/mL, yellow) and 0.1% ($\approx 10^3$ cells/mL, green) and below 0.1% (blue). Box plots are constructed with the upper and lower lines corresponding to the 25th and 75th percentiles; outliers are displayed as points.

when putative host abundance increased. This viral increase with host abundance was also observed in 98 other prokaryotic ASVs and their cooccurring viral species (Fig. 4; Fig. S6B and C). This result clearly indicates that frequency-dependent viral infection is prevalent in abundant prokaryotes at least between the detected virus-host pairs.

**Characterization of the virus-host interaction by host taxa.** The community of viruses showed a higher alpha-diversity than the community of prokaryotes (Fig. S2), and the cooccurrence analysis indicated one-to-many associations between the host and virus populations (median of 16 viral species per prokaryotic ASV). This suggests that one abundant prokaryotic ASV can interact with multiple viral species. Note that the numbers of cooccurring viral species were overestimated because each contig could be a partial genome fragment derived from the same viral genome (average completeness of mts-OBV contigs was 39%) (Table S1). However, the contigs classified into different genera (average of 8 gOTUs) often cooccurred with an ASV. Next, we characterized the "one-to-many" virus-host interaction network (i.e., how many viruses cooccurred with each ASV) with respect to their host taxa and host temporal dynamics.

The number of cooccurring viral species for prokaryotic ASVs was generally dependent on the predicted number of associated viruses determined via host prediction (Fig. S6D). For example, *Bacteroidetes* viruses (548 viruses) were the second most frequently observed viruses, and an average of 71.5 viruses cooccurred with *Bacteroidetes* ASVs (1 to 208 viruses per ASV, between 37 *Bacteroidetes* ASVs and 339 *Bacteroidetes* viruses). The number of cooccurring viruses could be overestimated because of the double count of cooccurring viruses between two cooccurring ASVs. If ASV-A and ASV-B cooccurred, the viruses cooccurring with ASV-A can also be included in the viruses cooccurring with ASV-B and vice versa. In fact, up to 16 ASV-ASV cooccurring pairs were detected for *Bacteroidetes*. In contrast, the taxa with less frequently detected

viruses (e.g., MGII, 38 viruses) had a smaller number of cooccurring populations (0 to 3 viruses per ASV) (Fig. S6D). Thus, the number of cooccurring viral species might be underestimated in these taxa because of host prediction limitations. In addition, it should be noted that there are two assumptions for our virus-host cooccurrence analysis: (i) viral host range is narrow as is common in isolated strains (59), and (ii) infectious virions do not persist a long time with low external virion inflow. The latter assumption is following the diel cycling and locality of viral production (34, 60, 61). Thus, it should be noted that exceptions for these assumptions, such as broad-host-range viruses (62), may cause misinterpretation from the cooccurrence analysis. Further studies are required for the quantification of viruses with such exceptional characteristics in the environment.

To confirm the validity of the cooccurrence analysis using the putative virus-host pairs derived from our genome-based host prediction, we compared the number of cooccurring pairs between them and nonhost pairs. In prokaryotic taxa that found many viruses (e.g., *Bacteroidetes* and *Alphaproteobacteria*), more frequent cooccurrences were observed between putative host taxa and their viruses than nonhost-like pairs (Fig. S6E to G). This result suggests that the genome-based host prediction assisted the connection of more plausible pairs. Further, to examine the robustness of our cooccurrence analysis, we converted the relative abundance to the absolute abundance for the samples with cell and virus-like particle (VLP) counts. In this analysis, at least 97 virus-host pairs were confirmed in absolute abundance levels (fig. S6H and I).

**Characterization of the virus-host interaction by host temporal dynamics.** Of note, SAR11 had relatively few cooccurring viral species even though there were more than 500 putative SAR11 viral species (Fig. S6D). SAR11 is often regarded as a *K*-strategist, which is believed to be resistant to viral infection (4), and the growth strategy may influence the cooccurrence dynamics with viruses. Next, we examined the number of cooccurring viruses among prokaryotic ASVs classified in the same taxa depending on the temporal dynamics to solve this issue.

It is difficult to determine the growth strategy of each prokaryotic ASV because maximum growth rate ($r$) and carry capacity ($K$) cannot be directly inferred from their monthly temporal dynamics (63). Thus, instead of assigning strategies to each ASV, we classified the ASVs into two groups according to their temporal dynamics (temporarily abundant or persistently abundant). The temporal dynamics of each prokaryotic ASV was assessed by the indexes that we introduced (see Materials and Methods). According to these, 13 ASVs were the most persistently dominant (i.e., *K*-like index of >12 and *r*-like index of <0.1). Among the 13 ASVs, 7 were classified as SAR11 (Fig. S7). Twenty-two of 57 ASVs belonging to the taxa previously predicted as *r*-strategists (i.e., *Flavobacteriaceae*, *Rhodobacteraceae*, *Vibrio*, and Marine Group II) were classified as temporarily abundant ASVs (*K*-like index of <3 and *r*-like index of >0.5; total of 33 ASVs) (Fig. S7). Generally, temporarily abundant ASVs, such as members of *Bacteroidetes*, showed many cooccurring viral species (Fig. S7). In contrast, persistently dominant ASVs of *Synechococcus* and SAR11 showed relatively few cooccurring viral species (Fig. S7). The most abundant ASV of *Synechococcus* (ASV8-1, making up 76.7% of all cyanobacterial reads) and SAR11 (ASV1-1, occupying 7 to 64% of all SAR11 reads of each month) showed 7 and 16 cooccurring viruses, respectively, even though 183 cyanoviruses and 500 SAR11 viruses were detected during the observation (Fig. S7).

The cyanoviruses and SAR11 viruses were as abundant as their putative host taxa (e.g., both cyanoviruses and cyanobacteria were dominant during the summer) (Fig. 2). The few cooccurrences of virus and host seem to be not reasonable. However, if a temporal shift of virus-host pairs occurred, cooccurrence analysis may fail to detect virus-host associations. Therefore, we compared the dynamics of the two dominant prokaryotic ASVs and viral species that did not cooccur with their predicted hosts. Representative sequences of ASV8-1 matched with the members of *Synechococcus* subcluster 5.1a with 100% identity. Among the 53 cyanoviral species that did not cooccur with any cyanobacterial ASV, 41 species were classified into two gOTUs (G14 [T7-like cyanosiphovirus] and G386 [T4-like cyanomyovirus]), which are known to infect subcluster 5.1a (e.g., *Synechococcus* sp. WH 8103, clade II), suggesting a plausible interaction between ASV8-1 and these

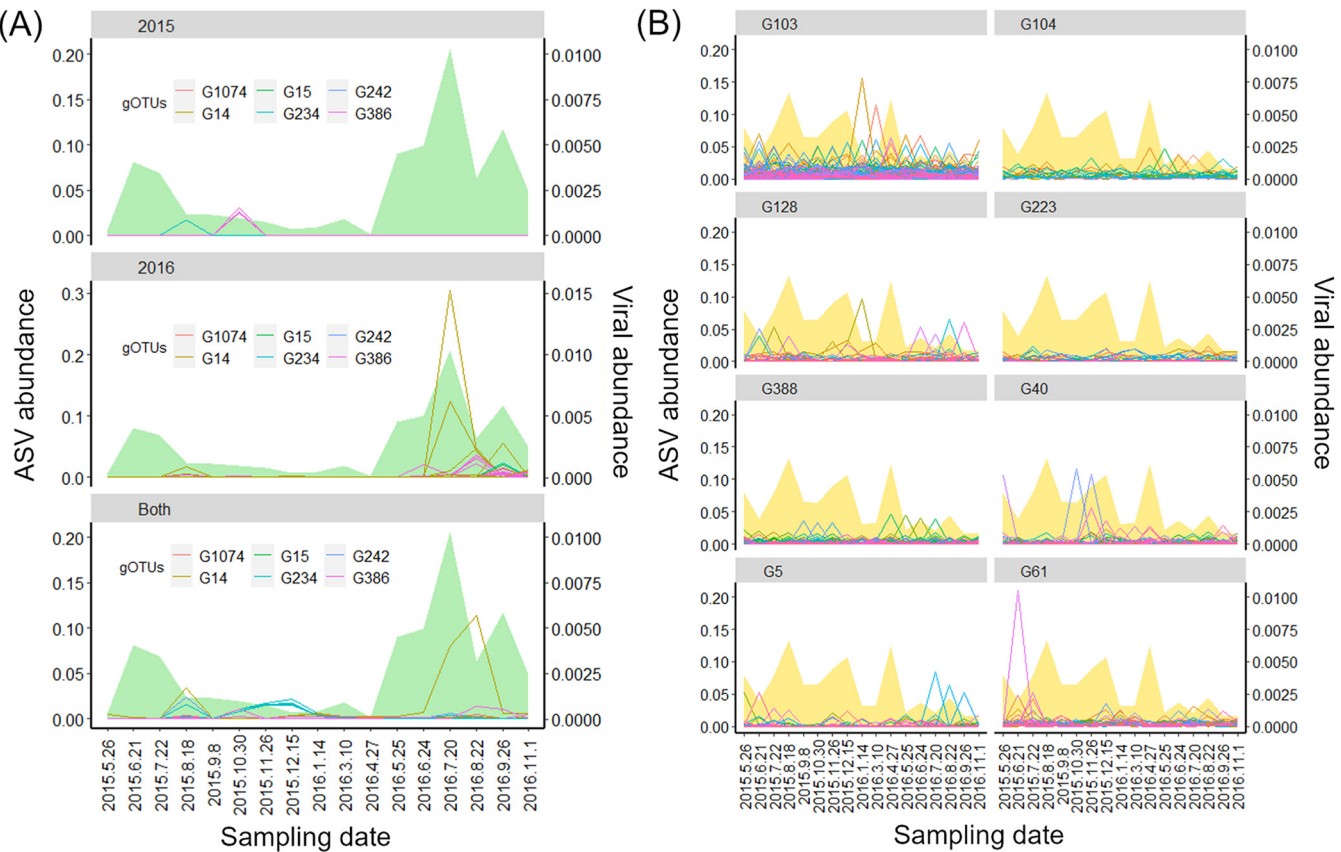

**FIG 5** Dynamics of the most dominant prokaryotic population (ASV1-1 and ASV8-1) and viruses that were predicted to infect these host taxa by host prediction analysis but did not cooccur with any ASV. (A) Dynamics of ASV8-1 that was classified into *Synechococcus* and 53 cyanoviruses that did not cooccur with cyanobacterial ASVs. The area chart represents the relative abundance of ASV8-1, and the lines represent viral contigs over time. The panels were separated by viral annual pattern (2015 type, 2016 type, and both years type; if the virus was more than five times more abundant in one year than in another year, the virus was defined as a year-specific virus). Colors represent the gOTU (genus) of the virus. (B) Dynamics of ASV1-1 that was classified into the SAR11 clade and 309 putative SAR11 viruses that did not cooccur with any SAR11 ASVs. The area chart represents the relative abundance of ASV1-1, and the lines represent viral contigs over time. The panels were separated based on the classified gOTUs of each virus.

viruses. ASV8-1 especially dominated during the summer (maximum of 8% and 21% of the prokaryotic community in June 2015 and July 2016, respectively) (Fig. 5A). Of these 53 viral species, the abundances of which also increased in summer, 4 were abundant only in 2015 (from 5 to greater than 170 times more abundant in 2015 than in 2016), and another 38 species were more abundant in 2016 (from 5 to greater than 300 times more abundant in 2016 than in 2015) (Fig. 5A). Similarly, ASV1-1 of SAR11 was always abundant (Fig. 5B), and SAR11 viruses occupied a major fraction of the viral community. However, abundant members of SAR11 viruses (309 contigs) were replaced in a relatively short time (a few months) (Fig. 5B). These results suggest that the host-virus interaction might have been underestimated in the cooccurrence analysis, and persistently dominant ASVs can also interact with multiple viruses based on their cell density. Considering a recent observation of infection dynamics of a lysogenic SAR11 virus, which did not show decreased host abundance even as virus abundance increased (51), the continuous dominance of the host with infection likely occurs versus assuming rare hosts produce a large amount of viruses. Although we cannot directly connect each ASV and its growth strategy, the prevalence of interactions between viruses and prokaryotes showing different temporal dynamics suggests that abundant prokaryotes are potentially infected by viruses according to their cell density regardless of their growth strategy.

Finally, we investigated whether the observed viruses, including those not statistically detected as cooccurring viruses with hosts (e.g., 53 cyanoviruses and 309 SAR11 viruses in Fig. 5), were also produced via increased contact frequency with hosts. To infer the contact frequency, we focused on single-nucleotide polymorphisms (SNPs) in viral

genomes. SNPs of closely related viral populations were previously observed in abundant viral populations, such as freshwater cyanoviruses (64) and marine viruses in other coastal areas (26). Because recent studies suggested that the majority of viruses observed in the virome were produced via diel and local viral-host interactions (28, 34, 38, 61), it likely indicates that multiple infection events may lead to the generation of mutations through DNA replication. We thus hypothesized frequent reproduction and mutations for abundant viruses with an increased contact frequency with their hosts. Therefore, SNPs from mts-OBV contigs with more than 10× coverage depth (2,356 contigs) were calculated. We observed increasing intrapopulation genetic diversity (SNPs quantified by average genomic entropy) as a function of overall viral population abundance regardless of their host taxa (Fig. S8). This suggests that the increase in contact frequency occurred not only in viruses showing clear cooccurrence with hosts (e.g., *Bacteroidetes* viruses) but also in viruses that showed no statistical detection of cooccurrence with hosts (e.g., SAR11 viruses). This result corroborates the notion that contact rate is the key parameter for viral reproduction regardless of whether they show a long-term cooccurrence pattern with their hosts.

**Ecological interpretation inferred from virus-host dynamics.** There are at least three possible mechanisms that could lead to shifts of the dominant viral species in the same host groups (Fig. 5). First, more closely related prokaryotic populations that cannot be differentiated by the 16S rRNA gene polymorphism could cooccur with viruses. Previous studies focusing on the polymorphism of internal transcribed spacer (ITS) sequences (ITS-ASV) in SAR11 and *Cyanobacteria* reported that ITS-ASV dynamics correlate more with viral dynamics (as inferred from T4-like viral marker genes) than 16S-ASV dynamics of these taxa (7, 29). Therefore, dynamics of more highly resolved populations (e.g., ITS-ASVs or whole-genome-sequence-based populations) might have synchronized with observed viral dynamics. Second, the temporal acquisition of host resistance or viral counterresistance, as often observed in culture model systems (65), may cause a shift of the dominant viral species. Third, the shifts of viruses can be interpreted as a result of the founder effect following host fluctuation via genetic drift (66). Seasonal fluctuation of the host population causes a bottleneck effect, and, therefore, the founder effect following the bottleneck effect enables the abundances of several viral species to equally increase. This was suggested as a mechanism of an incomplete selective sweep in the freshwater *Cyanobacteria* populations having different CRISPR-spacer genotypes (67). This scenario is more plausible between ASV8-1 and their viruses because ASV8-1 experienced clear seasonal fluctuation (Fig. 5A).

Altogether, we revealed that frequency-dependent infection occurrs in abundant prokaryotic populations according to cell density via "one-to-many" host-virus correspondences regardless of the host temporal dynamics. Although more direct evaluation of growth capacity is required for validation, this suggests "one-to-many" frequency-dependent infection can occur regardless of host growth strategy. Such "one-to-many" cooccurrences seem to support the validity of the previous study, which used cooccurrences between viruses for the host prediction method (68). One-to-many host-virus correspondences may suggest that a prokaryotic species can be attacked by multiple viruses with different infection strategies (e.g., different cell surface targets). This can cause difficulties in establishing complete resistance toward multiple coexisting viruses and sustaining continuous virus-host interaction in the environment. The inability to evolve complete resistance to many viral species in complex virus-host systems in the environment can contribute to the prevalence of frequency-dependent selection in abundant marine prokaryotes.

The comparison of monthly dynamics between prokaryotic and viral communities indicated concurrent seasonal shifts at the whole-community level. Concurrent seasonal shifts were also broadly observed between the corresponding virus and host pairs at the phylum or class level based on the host prediction analysis. We further statistically confirmed their cooccurrence via network analysis among abundant prokaryotic populations and their viruses regardless of the host taxa or temporal dynamics. These results likely suggest that abundant prokaryotes are broadly exposed to frequent viral infection

regardless of their taxa and growth strategy and indicate that lysis of the abundant prokaryotes via viral infection has a considerable impact on biogeochemical cycling and maintenance of prokaryotic community diversity. Further, these abundant prokaryotic populations should reflect actively growing members of the community because they became dominant even though they suffered frequent loss by viral lysis.

## MATERIALS AND METHODS

**Sampling and processing.** Seawater samples (4 l) were collected at a 5-m depth at the entrance of Osaka Bay (34°19′28″N, 135°7′15″E), Japan, between March 2015 and November 2016 at monthly intervals. Because the tide level and diel cycle can influence community composition, the sampling time was unified at 3 h before or after high tide in the daytime. Considering the climate of the sampling area, the seasons were defined as follows: March to May = spring, June to August = summer, September to November = autumn, and December to February = winter. Additionally, from June to July was considered the rainy season. Seawater was filtered through a 142-mm-diameter (3.0-$\mu$m-pore-size) polycarbonate membrane (Millipore, Billerica, MA) and then sequentially through 0.22-$\mu$m-pore-size Sterivex filtration units (SVGV010RS, EMD Millipore). After filtration, filtration units were directly stored at −80°C for subsequent DNA extraction. The filtrates were stored at 4°C before treatments. Water temperature and salinity were monitored using fixed water intake systems of the Research Institute of Environment, Agriculture, and Fisheries, Osaka prefecture. Nutrient concentrations ($NO_3$-N, $NO_2$-N, $NH_4$-N, $PO_4$-P, and $SiO_2$-Si) were measured by continuous flow analysis (BL TEC K.K., Japan). Prokaryotic cell and virus-like particles were enumerated using SYBR green and SYBR gold epifluorescence microscopy (69, 70).

**rRNA gene amplicon sequencing analysis.** For prokaryotic community analysis, DNA was extracted from the stored filtration units, as previously described (34, 71). Total 16S rRNA genes were amplified using a primer set based on the V3-V4 hypervariable region of prokaryotic 16S rRNA genes (72) with added overhang adapter sequences at each 5′ end according to the sample preparation guide (https://support.illumina.com/content/dam/illumina-support/documents/documentation/chemistry_documentation/16s/16s-metagenomic-library-prep-guide-15044223-b.pdf). Amplicons were sequenced using a MiSeq sequencing system and MiSeq V3 (2 × 300 bp) reagent kits (Illumina, San Diego, CA).

Paired-end 16S rRNA gene amplicon sequences were merged using VSEARCH with the "-M 1000" option (73). Merged reads containing ambiguous nucleotides (i.e., "N") were discarded. The remaining merged reads were clustered using VSEARCH to form operational taxonomic units (OTUs) at a 99% sequence identity threshold. Singleton OTUs were discarded. The representative sequences of the remaining OTUs were searched against the SILVA rRNA gene database (release 138) (74) to taxonomically annotate OTUs using SINA (75) at a 99% sequence identity threshold. To define abundant OTUs, we considered the reported minimum host cell density for effective viral infection ($\approx 10^4$ cells/mL) (58). Following typical coastal marine prokaryotic cell density ($\approx 10^6$ cells/mL) (76), we assumed prokaryotic cell density uniformity as $10^6$ cells/mL during the whole sampling period, and the cutoff value for abundant OTUs was set as 1% relative abundance ($10^6 \times 0.01$ [1%] = $10^4$ cells/mL).

To identify statistically relevant variants within abundant OTUs, we applied minimum entropy decomposition (MED) (11) as previously reported. All the sequences from each 99% OTU were aligned using MAFFT v7.123b (-retree 1 -maxiterate 0 -nofft -parttree) (77). Alignments of sequences containing positions with an entropy of >0.25 were decomposed, and decomposition continued until all positions had an entropy of <0.25. The minimum number of the most abundant sequence within each amplicon sequence variant (ASV) needed to exceed 50, and ASVs that did not exceed 1% of the parent OTU composition were discarded (7).

**Virome sequencing, assembly, classification, and calculation of relative abundance.** The filtrate containing viruses was concentrated via $FeCl_3$ precipitation (78) and purified using DNase and a CsCl density centrifugation step (79). The DNA was then extracted as previously described (80). We failed to obtain enough DNA for virome sequencing for one sample (February 2016), so the sample was removed from the analysis. Libraries were prepared using a Nextera XT DNA sample preparation kit (Illumina, San Diego, CA), according to the manufacturer's protocol, using 0.25 ng of viral DNA. Samples were sequenced using a MiSeq sequencing system and MiSeq V3 (2 × 300 bp) reagent kits (Illumina, San Diego, CA).

Viromes were individually assembled using SPAdes 3.9.1 with default $k$-mer lengths (81). Additionally, we used scaffolds of these assemblies (here referred to as contigs for simplicity). Only the virus-like contigs were extracted using VirSorter (categories 1, 2, and 3) (82). Circular contigs were determined as previously described (33). Contig sequences were clustered at 95% global average nucleotide identity with cd-hit-est (options -c 0.95 -G 1 -n 10 -mask NX; 549 redundant contigs were discarded) (83). A total of 5,226 monthly time series Osaka Bay viral (mts-OBV) contigs (>10 kb, 62 to 926 contigs/sample, including 202 circular contigs) were obtained. Genome completeness and quality of mts-OBV contigs were evaluated using checkV (v0.7.0) (84).

In addition, this assembly generated 181,131 short contigs (i.e., from 1 kb to 10 kb). Because the reliability of virus prediction tools decreases in shorter contigs, the abundance of these contigs was assessed based on the relative abundance of terminase large subunit genes (*terL*), as previously described (34). In total, 4,666 genes were detected as putative *terL* genes (i.e., genes with the best hits to PF03354.14, PF04466.12, PF03237.14, and PF05876.11). Fragments per kilobase per million (FPKM) values for putative *terL* genes were calculated using in-house ruby scripts (https://github.com/yosuken/CountMappedReads2).

The mts-OBV contigs with complete viral genomic sequence set collected in a previous study (33) were used for viral abundance estimation based on the read mapping. The complete viral genomic

sequence belonged to one of the following two categories: (i) 1,811 environmental viral genomes (EVGs; all are circularly assembled genomes, 45 were assembled in Osaka Bay in a previous study [33]) derived from marine virome studies and (ii) 2,429 reference viral genomes (RVGs) of cultured dsDNA viruses, which were collected previously (33). Genus-level genomic OTUs (gOTUs) were previously assigned for complete genomes based on the genomic similarity score ($S_G$) using ViPTree (85). For the mts-OBV contigs, if a sequence showed a high similarity to one of the complete genomes (with an $S_G$ of >0.15), the sequence was assigned to the gOTU of the most similar genome, as previously described (33, 34). Quality-controlled virome reads were obtained through quality control steps as previously described (33). These reads were mapped against the viral genomic sequence set using Bowtie2 software with the "–score-min L,0,-0.3" parameter (86). FPKM values were calculated using in-house ruby scripts, and relative abundances of each virus among the analyzed viruses were measured.

**Viral host prediction.** First, we assigned putative host groups based on genomic similarity with the viral genomic sequence set collected in a previous study (33). Among the 487 genera (gOTUs) of 2,429 cultured prokaryotic viral genomes in the previous classification, there were only two gOTUs that included viruses infecting two different host phyla (33). The exceptional gOTUs are G617 and G1038. G617 includes viruses that infect *Firmicutes* and *Deinococcus-Thermus*. Most of the members of G1038 infect *Enterobacteriales* in *Proteobacteria*, but a virus (*Staphylococcus* phage SA1) infects *Staphylococcus aureus* in *Firmicutes*. Thus, members in the same gOTU likely infect similar host groups, at least at a relatively broad taxonomic level as predicted in this study (phylum or class level in *Proteobacteria*). If mts-OBV contigs were classified into the same gOTU as the viruses with a known (via cultivation) or predicted (by genomic content [33]) host group, the host group was assigned to the contigs. We also compared similarity between mts-OBV contigs, the viral genomes deposited in Virus-Host DB (https://www.genome.jp/virushostdb/; as of October 2018), and recently reported isolates (87, 88).

For the viruses without assigned host groups via genomic similarity, we performed *in silico* host prediction based on the nucleotide sequence similarity between viruses and prokaryotes as follows according to previous studies (42, 89, 90). First, a total of 220,103 viral genomes/contigs derived from marine viromes were collected and used for the analysis (26, 33, 38, 43, 44) (Table S1 in the supplemental material). For the putative host genomes, we collected a total of 8,016 metagenome-assembled genomes (MAGs)/single amplified genomes (SAGs) from marine metagenomic or single-cell genomic studies (45–49). From Pachiadaki et al., we used only 1,040 high-quality SAG assemblies with ≥80% completion (49). To remove the contamination of virus-like contigs from the MAGs/SAGs, 14,967 contigs classified as viral-like sequences using VirSorter (categories 1, 2, and 3) (82) were discarded (Table S1). Details of each prediction method were reviewed previously (91).

**(i) CRISPR-spacer matching.** CRISPR-spacer sequences were predicted using the CRISPR Recognition Tool (92), and a total of 13,305 sequences were extracted from the analyzed 8,016 MAGs/SAGs. Detected spacer sequences and spacer sequences deposited in CIRSPRdb (93) were queried against viral genomes using the BLASTn-short function (94), where at least 95% identity over the whole spacer length and only 1 to 2 SNPs at the 5′ end of the sequence were allowed.

**(ii) tRNA matching.** tRNAs were recovered from the 8,016 MAGs/SAGs and viral genomes using ARAGORN with the "-t" option (95). A total of 213,939 and 31,439 tRNAs were recovered from MAGs/SAGs and viral genomes, respectively. The recovered prokaryotic and viral tRNAs with 111,385 tRNAs deposited in GtRNAdb (96) were compared using BLASTn (94), and only a perfect match (100% length and 100% sequence identity) was considered indicative of putative host-virus pairs.

**(iii) Nucleotide sequence homology of prokaryotic and viral genomes.** Viral genomes/contigs were queried against prokaryotic MAGs/SAGs and prokaryotic genomes in NCBI RefSeq (as of December 2019) using BLASTn (94). Only the best hits above 80% identity across the alignment with a length of ≥1,500 bp were considered indicative of host-virus pairs. For the prediction based on MAG/SAG contigs, we performed taxonomic validation of the matching contigs in MAGs/SAGs as previously described (42).

No inconsistent host prediction for each viral contig was observed between these three different methods. Viruses belonging to the same gOTU were assigned consistent host groups (33), with three exceptional gOTUs (G404 including putative alphaproteobacterial viruses and putative deltaproteobacterial viruses, G405 including putative *Bacteroidetes* viruses and putative *Marinimicrobia* viruses, and G495 including putative alphaproteobacterial viruses and putative *Bacteroidetes* viruses), which included viruses predicted to infect multiple host lineages. For the contigs assigned to the three gOTUs, genomic similarity scores among the same gOTU members were calculated, and the potential host of each contig was assigned based on the most similar genomes/contigs, which was annotated via host prediction.

**Statistical analyses.** Before statistical analyses, 16S rRNA amplicon reads were rarefied using the "vegan" package in R (20,803 reads per sample based on minimum sample size) (97). To examine within-sample alpha-diversity (Shannon diversity, evenness, and richness) and beta-diversity (Bray-Curtis similarity: 1 − Bray-Curtis dissimilarity for all possible pairwise combinations among all of sampling points), we used the vegan package in R (98). Mantel tests were performed using R and the vegan package (98) only on fully overlapping sets of data. Pairwise correlations between the estimated abundance of prokaryotic ASVs and viral contigs (with putative host information and exceeding FPKM values of >10 at least a month, 2,735 contigs) on fully overlapping sets of data were then determined via Spearman correlation ($\rho$>0.6, $P < 0.01$, $q < 0.05$), as implemented in the local similarity analysis program. (99, 100). Only the pairs supported by host prediction (e.g., *Bacteroidetes* and *Bacteroidetes* viruses) were considered cooccurring virus-host pairs. In the sanity check of the analysis (Figure S6E-G), pairs that were not supported by the host prediction were included. However, even though virus_A cooccurred with ASV_X, which was not supported by host prediction, virus_A cooccurred with another ASV_Y, which was supported by host prediction, and ASV_X and ASV_Y cooccurred; the pair between virus_A and ASV_X were

removed as a false positive. In absolute abundance-based analysis, the count data of prokaryotic cells and virus-like particles for the first 7 months were used because of the sample loss of other months. The absolute abundances of each ASV and virus were calculated as the multiplication of cell/virus-like particle counts and relative abundance. Network visualizations of correlation matrices were generated using igraph and ggnetwork (101, 102).

**Classification of temporal dynamics of ASVs.** We established indexes for the classification of temporal dynamics of each ASV by their monthly dynamics. For the analogy of the $r$ (intrinsic rate of natural increase) of each ASV, the maximum increase of the normalized relative rank (0 to 1) per month of each ASV was applied. Similarly, for the analogy of $K$ (carrying capacity) for each ASV, the length of the continuously abundant month (>0.1% relative abundance, 1 to 18 months) of each ASV was applied. Although these indices are not the same as the original definitions of $r$ and $K$, we refer to these two indices as an $r$-like index and a $K$-like index, respectively, for simplicity.

**Detection of SNPs.** Reads were mapped to viral contigs using Bowtie2 with a "–score-min L,0,-0.3" (86), and the resulting alignment files were converted to BAM format and sorted using SAMtools (103). The average genome entropy of the contigs, which exceeded more than $10\times$ coverage each month, was computed using DiversiTools (http://josephhughes.github.io/DiversiTools/).

**Data availability.** Sequences obtained from the observations were deposited at the DNA Data Bank of Japan (DDBJ) under project number PRJDB10879. Raw sequence reads can be found under accession numbers DRX260081 to DRX260115, and assemblies of viromes can be found under BioSample ID SAMD00279559.

## SUPPLEMENTAL MATERIAL

Supplemental material is available online only.
**FIG S1**, PNG file, 0.2 MB.
**FIG S2**, PNG file, 0.8 MB.
**FIG S3**, PNG file, 0.1 MB.
**FIG S4**, PNG file, 0.5 MB.
**FIG S5**, PNG file, 0.6 MB.
**FIG S6**, PDF file, 0.5 MB.
**FIG S7**, PNG file, 0.6 MB.
**FIG S8**, PNG file, 0.3 MB.
**TABLE S1**, XLSX file, 7.8 MB.
**TABLE S2**, XLSX file, 0.01 MB.

## ACKNOWLEDGMENTS

K.T. would like to thank Ryoma Kamikawa, Shigeki Sawayama, and Daichi Morimoto at Kyoto University for their technical comments and preparation of the manuscript. K.T. would also like to thank Hisashi Endo, Florian Prodinger, Hiroaki Takebe, Kentaro Fujiwara, and Tatsuhiro Isozaki at Kyoto University and Keizo Nagasaki at Kochi University for very useful discussions. We thank the Research Institute of Environment, Agriculture, and Fisheries, Osaka prefecture, for supporting the sampling and Sakie Tanaka for continuous flow analysis. Computational work was supported by the Super Computer System, Institute for Chemical Research, Kyoto University. This study was supported by Grants-in-Aid for Scientific Research KAKENHI (numbers 17H03850 and 21H05057) and Challenging Exploratory Research (number 26660171) from the Japan Society for the Promotion of Science (JSPS), Canon Foundation (number 203143100025), JSPS Scientific Research on Innovative Areas (number 16H06437), and the Bilateral Open Partnership Joint Research Project (Japan-Lithuania Research Cooperative Program) "Research on prediction of environmental change in Baltic Sea based on comprehensive metagenomic analysis of microbial viruses."

K.T. performed the experiments, analysis, and preparation of the manuscript. N.O.-H. and H.W. performed sampling, experiments, and analysis. K.Y. contributed to sampling and experiments. Y.N. and H.O. contributed to the analysis, discussion, and preparation of the manuscript. T.Y. contributed to the research design, discussion, manuscript revision, and overall support for this study.

We declare that the research was conducted in the absence of any commercial or financial relationships that could be construed as a potential conflict of interest.

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
