## [Reviewer comments · mSystems]

Prevalence of viral frequency-dependent infection in coastal marine prokaryotes revealed using monthly time series virome analysis

Kento Tominaga, Nana Ogawa-Haruki, Yosuke Nishimura, Hiroyasu Watai, Keigo Yamamoto, Hiroyuki Ogata, and Takashi Yoshida

Corresponding Author(s): Takashi Yoshida, Kyoto Daigaku

Review Timeline:

Submission Date:	September 26, 2022
Editorial Decision:	November 16, 2022
Revision Received:	December 23, 2022
Accepted:	December 29, 2022

Editor: Michael Rappe

Reviewer(s): The reviewers have opted to remain anonymous.

Transaction Report:

DOI: <https://doi.org/10.1128/msystems.00931-22>

November 16, 2022

Prof. Takashi Yoshida
Kyoto Daigaku
Graduate School of Agriculture
Sakyo-ku, Kitashirakawa-Oiwake
Kyoto, Kyoto 606-8502
Japan

Re: mSystems00931-22 (Prevalence of viral frequency-dependent infection in coastal marine prokaryotes revealed using monthly time series virome analysis)

Dear Prof. Takashi Yoshida:

Thank you for submitting your manuscript to mSystems. We have completed our review and I am pleased to inform you that, in principle, we expect to accept it for publication in mSystems. However, acceptance will not be final until you have adequately addressed the reviewer comments.

Preparing Revision Guidelines

Sincerely,

Michael Rappe

Editor, mSystems

Journals Department

Reviewer comments:

Reviewer #1 (Comments for the Author):

General comments

In this study by Kento et al, the authors examine the presence and abundance patterns of different virus-host pairs over a 2 year time series. They find that regardless of a bacteria's temporal pattern, the abundance of its viruses rises when the host is more abundant and present, supporting frequency-dependence models of viral lysis in eutrophic systems. This work untangles virus-host interactions at a higher resolution by including diverse members of the marine community compared to single host-pair studies. The findings are important by numerically demonstrating the frequency-dependence of viruses on their hosts' presence and abundance, regardless of the temporal stability of that host (i.e. the rise in a host leads to the rise in its viruses, even if that virus is relatively constant in the community). My main concerns are the host predictions and their inference of growth strategy/life history strategy based on temporal presence.

For the host prediction, in the Methods section, the impression is that contigs belonging to the same genus (gOTU) are assigned the same host, even if only one contig's host could be predicted (lines 524-527). They do not mention whether these host-associations were always consistent (perhaps two cultivated viruses within a gOTU have different hosts. How was this handled?). For the other gOTUs, they use tRNA, CRISPR, and genomic similarity approaches, which all can have different results, but they only reported the result from one method in Supplementary Dataset 5. They do not specify how they handled inconsistent predictions (was one method given preference as more reliable than the other? Did these methods all align for a single contig? Let alone for all the contigs within a gOTU?) At the gOTU-level, they found a 3 cases where the host of the contigs in the same gOTU differed, but didn't elaborate on the taxonomic level that these hosts differed (same class? same phylum?), perhaps these predictions are the true host of each contig but the genus of viruses infects a broad group of bacteria. The lack of details on host predictions for each contig and each gOTU makes it hard to assess the confidence of these virus-host associations for the study overall. They need to provide a table with each contig's results for the different methods (or explain the hierarchy used to determine the host by the different methods) and the consensus host, in addition to a table with each gOTU and the host of each contig within that OTU (including whether it was known or unknown). For instance, Supplemental Dataset 5 shows each contig's assignment but only reports the assignment from a single method and does not show the results of each method (tRNA, CRISPR, Blastn). This makes it unclear the consistency of the host prediction for the contig, let alone the contigs within a gOTUs.

The authors distinguish r-like strategies from K-like strategists based on the presence and relative abundance of the prokaryote over the 18 months, with K-like strategists being present in at least 0.1% over 18 months (lines 591-599). They acknowledge that monthly timepoints cannot directly determine growth strategy, but continue to refer to temporary or persistent presence as "growth strategy". The comparison viral communities infecting "stable hosts" versus "temporary hosts" is interesting in to compare. However, the authors do not provide evidence that a prokaryote's stability has been linked to growth rate. While intuitively this makes sense that more persistent members grow slower, this claim needs some direct evidence; otherwise, I recommend they refer to these host groups as "persistent" and "temporary" members of the communities and focus the discussion of prokaryote stability as it relates to the diversity, presence, and abundance of viruses that putatively infect them.

My minor comments mostly concerned clarification that can be addressed with some rewording.

Figure 1 - x-axis says 1-18, but unclear which numbers correspond to which months and season, making it difficult to follow the discussion. It would also be helpful if the authors could add a color strip under the x-axis showing the general timepoints of different seasons (e.g. blue for rainy season - brown for dry season)

Figure 3 - A little confused on how to read this network since some of the numbers don't match what is shown. For instance, for MGII, it says there are 6 ASVs and 2 viruses, but there are only 2 boxes (prokaryote ASVs) and 6 gray checks (viral contigs)? shown Was this a typo or am I misreading this?

Supplemental: After Figure S12, the numbers went back to S10 and S11 but should be S13 and S14

line 119 - why 18 samples and not 24? when were the samples collected? in daylight? This is elaborated in the methods but is crucial to following the discussion, considering prokaryotes and viruses experience diel cycling.

line 132 - were these contigs de-replicated to represent populations? (I saw in the Methods that they were, but this needs to be made clearer). How was completeness assessed beyond circularity, particularly as they later state that most contigs had only

39% completeness according to CheckV? Later on in lines 151-153, it says only the "complete genomes" were investigated, but it's unclear how this relates to the 5,226 contigs called the mts-OBV with only 202 circular genomes. Please clarify.

lines 169 - 181 : It would be helpful if this section more explicitly links community variation to the season. Was the variation highest in the rainy season? Did this rainy-dry oscillation match what was found in the SPOT study?

lines 253-255: Very interesting that ASVs within an OTU varied in their abundance patterns and am glad this is mentioned. It would be useful if elaborate on the classifications of OTU2 and OTU14 and speculate how the ASV variation may relate to the biology of these taxa.

line 248 - 261 : A little confused on the difference between the detection of virus-host pairs and correlations between ASVs and viruses (Supplementary Figure 6) versus the analysis of "whether the viruses were abundant when the host was abundant" of lines 265-267. Is this not the same correlation? Might be clearer to say that the virus's relative abundance amongst the viral community was compared to the dominance of the host among the prokaryotic community.

line 352: The use of the term "shift" instead of "switch" would be a little more accurate to describe the change in viruses infecting a host. The word "switch" can imply that suddenly two viruses are swapping which hosts they infect but they are both still present, which is not necessarily what is happening.

line 389 : by "ten coverages" do you mean 10x coverage depth?

Line 397-398: Again, saying virus-host pair "switch" implies an exchange between which viruses infect which ASVs (e.g. virus_9 infects ASV_2 and virus_7 infects ASV_5 but not virus_9 infects ASV_5 and virus_2 infects ASV_2). It would be clearer to say "There are at least three possible mechanisms that could lead to shifts in which groups (or populations) of viruses infect a host" or something along those lines.

Line 420: add "can be" after "may suggest a prokaryotic species..." Would then read "may suggest a prokaryotic species can be attacked by multiple"

Line 424: Unclear what the "mechanism" is. Do you mean "the inability of abundant marine prokaryotes to evolve complete resistance to viral infections contributes to their experiencing frequency-dependent selection" ?

Line 503: Did all of the short contigs encode TerL?

Lines 492 - 500: Unclear how viral contigs were distinguished from cellular contigs in the mts-OBV dataset. Was it just CheckV? If so, how was the output used to determine viral sequences? Just the default output or additional quality filtering?

line 513: Which database(s) did the reference genomes come from?

line 564: How divergent were the different hosts of the contigs within those 3 gOTUs that did not have consistent host predictions? This gives the impression that using a genus assignment to associate a contig with a host is not always accurate. Should provide a table overview of gOTU host prediction methods and results that lead to consensus host assignment.

Reviewer #3 (Comments for the Author):

The manuscript investigates a very interesting research question and describes an impressively extensive body of work. The sampling approach and lab methods are appropriate. The effort and diligence that went into the data processing and statistical analysis is remarkable. Only few methodological steps would need a more explanation, mainly how the prok ASV abundances were determined and normalized. In the results and discussion section I was a few times surprised by the choice of presented data/ figures. Some times instead of presenting the concluding stats test e.g. bar graphs are shown which the reader has to analyze themselves. Some times the interesting results are in the suppl material and I would suggest reconsidering the figure selection. The final section about the r and k strategists is based on a lot of assumptions and very speculative. At this stage I think this part would need more convincing discussion. Please find detailed comments attached. Congratulations to this impressive body of work.

General comments

In this study by Kento et al, the authors examine the presence and abundance patterns of different virus-host pairs over a 2 year time series. They find that regardless of a bacteria's temporal pattern, the abundance of its viruses rises when the host is more abundant and present, supporting frequency-dependence models of viral lysis in eutrophic systems. This work untangles virus-host interactions at a higher resolution by including diverse members of the marine community compared to single host-pair studies. The findings are important by numerically demonstrating the frequency-dependence of viruses on their hosts' presence and abundance, regardless of the temporal stability of that host (i.e. the rise in a host leads to the rise in its viruses, even if that virus is relatively constant in the community). My main concerns are the host predictions and their inference of growth strategy/life history strategy based on temporal presence.

For the host prediction, in the Methods section, the impression is that contigs belonging to the same genus (gOTU) are assigned the same host, even if only one contig's host could be predicted (lines 524-527). They do not mention whether these host-associations were always consistent (perhaps two cultivated viruses within a gOTU have different hosts. How was this handled?). For the other gOTUs, they use tRNA, CRISPR, and genomic similarity approaches, which all can have different results, but they only reported the result from one method in Supplementary Dataset 5. They do not specify how they handled inconsistent predictions (was one method given preference as more reliable than the other? Did these methods all align for a single contig? Let alone for all the contigs within a gOTU?) At the gOTU-level, they found a 3 cases where the host of the contigs in the same gOTU differed, but didn't elaborate on the taxonomic level that these hosts differed (same class? same phylum?), perhaps these predictions are the true host of each contig but the genus of viruses infects a broad group of bacteria. The lack of details on host predictions for each contig and each gOTU makes it hard to assess the confidence of these virus-host associations for the study overall. They need to provide a table with each contig's results for the different methods (or explain the hierarchy used to determine the host by the different methods) and the consensus host, in addition to a table with each gOTU and the host of each contig within that OTU (including whether it was known or unknown). For instance, Supplemental Dataset 5 shows each contig's assignment but only reports the assignment from a single method and does not show the results of each method (tRNA, CRISPR, Blastn). This makes it unclear the consistency of the host prediction for the contig, let alone the contigs within a gOTUs.

The authors distinguish r-like strategies from K-like strategists based on the presence and relative abundance of the prokaryote over the 18 months, with K-like strategists being present in at least 0.1% over 18 months (lines 591-599). They acknowledge that monthly timepoints cannot directly determine growth strategy, but continue to refer to temporary or persistent presence as "growth strategy". The comparison viral communities infecting "stable hosts" versus "temporary hosts" is interesting in to compare. However, the authors do not provide evidence that a prokaryote's stability has been linked to growth rate. While intuitively this makes sense that more persistent members grow slower, this claim needs some direct

evidence; otherwise, I recommend they refer to these host groups as "persistent" and "temporary" members of the communities and focus the discussion of prokaryote stability as it relates to the diversity, presence, and abundance of viruses that putatively infect them.

My minor comments mostly concerned clarification that can be addressed with some rewording.

Figure 1 - x-axis says 1-18, but unclear which numbers correspond to which months and season, making it difficult to follow the discussion. It would also be helpful if the authors could add a color strip under the x-axis showing the general timepoints of different seasons (e.g. blue for rainy season - brown for dry season)

Figure 3 - A little confused on how to read this network since some of the numbers don't match what is shown. For instance, for MGII, it says there are 6 ASVs and 2 viruses, but there are only 2 boxes (prokaryote ASVs) and 6 gray checks (viral contigs)? shown Was this a typo or am I misreading this?

Supplemental: After Figure S12, the numbers went back to S10 and S11 but should be S13 and S14

line 119 - why 18 samples and not 24? when were the samples collected? in daylight? This is elaborated in the methods but is crucial to following the discussion, considering prokaryotes and viruses experience diel cycling.

line 132 - were these contigs de-replicated to represent populations? (I saw in the Methods that they were, but this needs to be made clearer). How was completeness assessed beyond circularity, particularly as they later state that most contigs had only 39% completeness according to CheckV? Later on in lines 151-153, it says only the "complete genomes" were investigated, but it's unclear how this relates to the 5,226 contigs called the mts-OBV with only 202 circular genomes. Please clarify.

lines 169 - 181 : It would be helpful if this section more explicitly links community variation to the season. Was the variation highest in the rainy season? Did this rainy-dry oscillation match what was found in the SPOT study?

lines 253-255: Very interesting that ASVs within an OTU varied in their abundance patterns and am glad this is mentioned. It would be useful if elaborate on the classifications of OTU2 and OTU14 and speculate how the ASV variation may relate to the biology of these taxa.

line 248 - 261 : A little confused on the difference between the detection of virus-host pairs and correlations between ASVs and viruses (Supplementary Figure 6) versus the analysis of "whether the viruses were abundant when the host was abundant" of lines 265-267. Is this not the same correlation? Might be clearer to say that the virus's relative abundance amongst the

viral community was compared to the dominance of the host among the prokaryotic community.

line 352: The use of the term "shift" instead of "switch" would be a little more accurate to describe the change in viruses infecting a host. The word "switch" can imply that suddenly two viruses are swapping which hosts they infect but they are both still present, which is not necessarily what is happening.

line 389 : by "ten coverages" do you mean 10x coverage depth?

Line 397-398: Again, saying virus-host pair "switch" implies an exchange between which viruses infect which ASVs (e.g. virus_9 infects ASV_2 and virus_7 infects ASV_5 but not virus_9 infects ASV_5 and virus_2 infects ASV_2). It would be clearer to say "There are at least three possible mechanisms that could lead to shifts in which groups (or populations) of viruses infect a host" or something along those lines.

Line 420: add "can be" after "may suggest a prokaryotic species..." Would then read "may suggest a prokaryotic species can be attacked by multiple"

Line 424: Unclear what the "mechanism" is. Do you mean "the inability of abundant marine prokaryotes to evolve complete resistance to viral infections contributes to their experiencing frequency-dependent selection" ?

Line 503: Did all of the short contigs encode TerL?

Lines 492 - 500: Unclear how viral contigs were distinguished from cellular contigs in the mts-OBV dataset. Was it just CheckV? If so, how was the output used to determine viral sequences? Just the default output or additional quality filtering?

line 513: Which database(s) did the reference genomes come from?

line 564: How divergent were the different hosts of the contigs within those 3 gOTUs that did not have consistent host predictions? This gives the impression that using a genus assignment to associate a contig with a host is not always accurate. Should provide a table overview of gOTU host prediction methods and results that lead to consensus host assignment.

General comments

In this study by Kento et al, the authors examine the presence and abundance patterns of different virus-host pairs over a 2 year time series. They find that regardless of a bacteria's temporal pattern, the abundance of its viruses rises when the host is more abundant and present, supporting frequency-dependence models of viral lysis in eutrophic systems. This work untangles virus-host interactions at a higher resolution by including diverse members of the marine community compared to single host-pair studies. The findings are important by numerically demonstrating the frequency-dependence of viruses on their hosts' presence and abundance, regardless of the temporal stability of that host (i.e. the rise in a host leads to the rise in its viruses, even if that virus is relatively constant in the community). My main concerns are the host predictions and their inference of growth strategy/life history strategy based on temporal presence.

→ We would like to thank the reviewer for the meticulous review and derived comments and suggestions. Your comments give a lot of help in improving the manuscript. We have carefully gone over all the points raised and revised the manuscript accordingly. Please see the following response to each of the points raised in the revision.

For the host prediction, in the Methods section, the impression is that contigs belonging to the same genus (gOTU) are assigned the same host, even if only one contig's host could be predicted (lines 524-527). They do not mention whether these host associations were always consistent (perhaps two cultivated viruses within a gOTU have different hosts. How was this handled?).

→ Thank you for the important question. In the previous classification by Nishimura et al *mSphere* 2017, there were only two genera (gOTUs) which included viruses infecting different host phyla among 487 gOTUs of 2,429 cultured prokaryotic viral genomes. This suggests that members in the same gOTU likely infect similar host groups in at least at a relatively broad taxonomic level as in this study (phylum or class level in proteobacteria). Thus, we assigned the same host group to each gOTU if at least one contig of the host could be predicted. We added sentences to clarify it (please see LL498-505.).

For the other gOTUs, they use tRNA, CRISPR, and genomic similarity approaches, which all can have different results, but they only reported the result from one method in Supplementary Dataset 5. They do not specify how they handled inconsistent predictions (was one method given preference as more reliable than the other? Did these methods all align for a single contig? Let alone for all the contigs within a gOTU?)

At the gOTU-level, they found a 3 cases where the host of the contigs in the same gOTU differed, but didn't elaborate on the taxonomic level that these hosts differed (same class? same phylum?), perhaps these predictions are the true host of each contig but the genus of viruses infects a broad group of bacteria. The lack of details on host predictions for each contig and each gOTU makes it hard to assess the confidence of these virus-host associations for the study overall. They need to provide a table with each contig's results for the different methods (or explain the hierarchy used to determine the host by the different methods) and the consensus host, in addition to a table with each gOTU and the host of each contig within that OTU (including whether it was known or unknown). For instance, Supplemental Dataset 5 shows each contig's assignment but only reports the assignment from a single method and does not show the results of each method (tRNA, CRISPR, Blastn). This makes it unclear the consistency of the host prediction for the contig, let alone the contigs within a gOTUs.

→ Thank you for important questions and suggestions about host predictions by host-virus sequence similarity.

First, there was no case where multiple prediction methods (tRNA, CRISPR, and genomic similarity) predicted different hosts for the same contig. This indicates Supplementary Table S4 shows the whole result of host prediction analysis for each contig. Second, we put the highest priority on gOTU-based prediction because it shows good performance, as mentioned above. In addition, the host of the referenced viruses by gOTU method is based on data from culture experiments or previous predictions based on multiple circumstantial evidence, which appears to be more reliable than short sequence similarity with the host sequence used in other methods. Finally, there were three gOTUs for which no consensus host taxa within a single gOTU by 3 methods (tRNA, CRISPR, and genomic similarity). For each contig in the gOTUs, the same host group was assigned as the contig with the highest genomic similarity among the contigs which detected significant similarity with host sequence. We added this information in the method section of host prediction (LL540-548).

The authors distinguish r-like strategies from K-like strategists based on the presence and relative abundance of the prokaryote over the 18 months, with K-like strategists being present in at least 0.1% over 18 months (lines 591–599). They acknowledge that monthly timepoints cannot directly determine growth strategy, but continue to refer to temporary or persistent presence as “growth strategy”. The comparison viral communities infecting “stable hosts” versus “temporary hosts” is interesting in to compare. However, the authors do not provide evidence that a prokaryote’s stability has been linked to growth rate. While intuitively this makes sense that more persistent members grow slower, this claim needs some direct evidence; otherwise, I recommend they refer to these host groups as “persistent” and “temporary” members of the communities and focus the discussion of prokaryote stability as it relates to the diversity, presence, and abundance of viruses that putatively infect them.

→ Thank you for the suggestion. According to the comment, we changed the term “growth strategy” to “temporal dynamics” throughout the manuscript as recommended except for some discussion. These discussions about growth strategy are also retained with the excuse of shortages of direct evidence (LL351–354, LL391–393).

My minor comments mostly concerned clarification that can be addressed with some rewording.

Figure 1 – x-axis says 1–18, but unclear which numbers correspond to which months and season, making it difficult to follow the discussion. It would also be helpful if the authors could add a color strip under the x-axis showing the general timepoints of different seasons (e.g. blue for rainy season – brown for dry season)

→ Thank you for the helpful suggestion. We added information on typical seasons in Japan in the x-axis by color. Also, we added a definition of the seasons in the method section. Please see LL418–421.

Figure 3 – A little confused on how to read this network since some of the numbers don’t match what is shown. For instance, for MGII, it says there are 6 ASVs and 2 viruses, but there are only 2 boxes (prokaryote ASVs) and 6 gray checks (viral contigs)? shown Was this a typo or am I misreading this?

→ Sorry for the confusing text. This was just a typographical error. “2ASVs and 6 viruses” is the correct information. We revised the figure. Please see the revised Figure 3.

Supplemental: After Figure S12, the numbers went back to S10 and S11 but should be S13 and S14

→ Thank you for pointing out the mistake. We revised the figure numbers of the supplement datasets.

line 119 – why 18 samples and not 24? when were the samples collected? in daylight? This is elaborated in the methods but is crucial to following the discussion, considering prokaryotes and viruses experience diel cycling.

→ Thank you for the questions. We had planned to obtain samples for at least 12 months in this study to observe the seasonal dynamics of viruses and prokaryotes. The termination at 18 months was not a biological reason. It was more idealistic to observe for 2 years or more, but it was difficult due to staffing and research funding.

Since the tide level also influences the community composition in this station, the sampling time was unified at 3 h before or after high tide in the daytime. We added the information in the method section (LL416–418) and briefly in the result section (LL107–108).

line 132 – were these contigs de-replicated to represent populations? (I saw in the Methods that they were, but this needs to be made clearer). How was completeness assessed beyond circularity, particularly as they later state that most contigs had only 39% completeness according to CheckV?

→ We added the description of the “de-replication” step in the result section (See L118).

The completeness of viral contigs was assessed by CheckV. We did not set the cutoff of completeness except for size selection (>10 kb). We noticed the number of viral species can be overestimated because of the low completeness. Please see the excuse about it in LL267–270.

Later on in lines 151–153, it says only the “complete genomes” were investigated, but it’s unclear how this relates to the 5,226 contigs called the mts-OBV with only 202 circular genomes. Please clarify.

→ Sorry for the misunderstanding description. We used mts-OBVs “including” complete genomes. We corrected the sentence to make it clearer. Please corrected sentences in LL142-143.

lines 169 – 181 : It would be helpful if this section more explicitly links community variation to the season. Was the variation highest in the rainy season? Did this rainy-dry oscillation match what was found in the SPOT study?

Thank you for the insightful suggestion. Especially the community variation is generally larger (low similarity) between the autumn and spring samples. To show the trend, we added a Figure as S3. Also please see LL167-169. Japan is a humid climate throughout the year, but June to July is the rainiest season, so we called it the rainy season.

lines 253-255: Very interesting that ASVs within an OTU varied in their abundance patterns and am glad this is mentioned. It would be useful if elaborate on the classifications of OTU2 and OTU14 and speculate how the ASV variation may relate to the biology of these taxa.

Thank you for the suggestion. We added the taxonomic classifications of these ASVs and described that their parent OTUs showed stable dominant patterns (LL234-237).

line 248 – 261 : A little confused on the difference between the detection of virus-host pairs and correlations between ASVs and viruses (Supplementary Figure 6) versus the analysis of “whether the viruses were abundant when the host was abundant” of lines 265-267. Is this not the same correlation? Might be clearer to say that the virus’s relative abundance amongst the viral community was compared to the dominance of the host among the prokaryotic community.

→ Sorry for the unclear sentences and thank you for the suggestions for the correction. We corrected the sentences in LL 246-250 to make them clear according to the comments.

line 352: The use of the term “shift” instead of “switch” would be a little more accurate to describe the change in viruses infecting a host. The word “switch” can imply that suddenly two viruses are swapping which hosts they infect but

they are both still present, which is not necessarily what is happening.

Line 397–398: Again, saying virus–host pair “switch” implies an exchange between which viruses infect which ASVs (e.g. virus_9 infects ASV_2 and virus_7 infects ASV_5 but not virus_9 infects ASV_5 and virus_2 infects ASV_2). It would be clearer to say “There are at least three possible mechanisms that could leads to shifts in which groups (or populations) of viruses infect a host” or something along those lines.

→ Thank you for these corrections. “Shift” must be the more appropriate word in these sentences. We corrected L329, L372, and L380.

line 389 : by “ten coverages” do you mean 10x coverage depth?

→ Thank you for the correction. “10x coverage depth” is more clear expression. We corrected L364.

Line 420: add “can be” after “may suggest a prokaryotic species...” Would then read “may suggest a prokaryotic species can be attacked by multiple”

→ Thank you for the suggestion. According to the comments, we add “can be” before “attacked ~” in L395.

Line 424: Unclear what the “mechanism” is. Do you mean “the inability of abundant marine prokaryotes to evolve complete resistance to viral infections contributes to their experiencing frequency–dependent selection” ?

→ Thank you for the suggestion. We corrected the sentence according to the comments to make it clear. Please see LL398–340

Line 503: Did all of the short contigs encode TerL?

→ No. Since the predictability of virus–prediction tools (e.g., VirSorter) can decrease in shorter contigs, we only focused on the conserved marker gene. Other short contigs were not considered in this study. We added the reason in the corresponding sentence (L473–475).

Lines 492 – 500: Unclear how viral contigs were distinguished from cellular contigs in the mts–OBV dataset. Was it just CheckV? If so, how was the output used to determine viral sequences? Just the default output or additional quality filtering?

→ Sorry, we unintentionally forgot to describe the screening of virus–like

contigs in the method section. Thanks for noticing that. We selected virus-like contigs by filtering via VirSorter. We added the step in the method section L466.

line 513: Which database(s) did the reference genomes come from?

→Thank you for the comments. The RVG data set was collected in a previous study (Nishimura et al , mSphere, 2017). We added the information with the citation (L486).

line 564: How divergent were the different hosts of the contigs within those 3 gOTUs that did not have consistent host predictions? This gives the impression that using a genus assignment to associate a contig with a host is not always accurate. Should provide a table overview of gOTU host prediction methods and results that lead to consensus host assignment.

→Thank you for the comments. We summarized these unmatched examples. Please see LL 541–544.

Reviewer #3 (Comments for the Author):

The manuscript investigates a very interesting research question and describes an impressively extensive body of work. The sampling approach and lab methods are appropriate. The effort and diligence that went into the data processing and statistical analysis is remarkable. Only few methodological steps would need a more explanation, mainly how the prok ASV abundances were determined and normalized. In the results and discussion section I was a few times surprised by the choice of presented data/ figures. Some times instead of presenting the concluding stats test e.g. bar graphs are shown which the reader has to analyze themselves. Some times the interesting results are in the suppl material and I would suggest reconsidering the figure selection. The final section about the r and k strategists is based on a lot of assumptions and very speculative. At this stage I think this part would need more convincing discussion. Please find detailed comments attached. Congratulations to this impressive body of work.

We would like to thank the reviewer for the meticulous review and derived comments, suggestions, and encouragement. We revised manuscripts according to the comments from you, and another reviewer.

According to your and another reviewer's comments, we have weakened the claim about the growth strategy. Generally, we discussed "stable hosts" and "temporary hosts" and stopped directly discussing their growth strategy. We pointed out the possibility that these can also be related to r-/k- but we also pointed out the need for more investigation. We also moved several supplement data to the main data according to the suggestion. For the retained one, please see our response for the corresponding figures.

138-142 does that mean that on average the remaining 60% of reads would map to the 25% of contigs that were not previously reported? Are the mts-OBV from this study not the same as the previously reported ones? Please clarify.

⇒ Sorry for the unclear expressions. The remaining 60 % of reads were not mapped to the any of contigs we used for the mapping analysis (mts-OBV contigs and previously published genomes). Presumably, most unmapped reads can be mapped on the shorter contigs (<10 kb) or on contigs that were removed as contamination by non-viral sequences.

The mts-OBV is not the same contig as the previously reported contigs (mts- is short for monthly time series).

We changed "or" to "and" in L129 to make them clear.

156-164 Comparing diversity values between viral and cellular entities on different taxonomic resolutions does not seem the best approach. Maybe reconsider.

⇒ Thank you for the insightful suggestion. We noticed the difficulty in a direct comparison of the diversity of viral and prokaryotic communities as a comparison among the taxa in the same domain. However, we believe it is important to describe their diversity at least in a common taxonomic resolution for each group. This will help the ecological understanding of the relationship between each community, at least in terms of human interpretation. Although we confirmed the statistical significance of alpha diversity between communities, the significance can be meaningless as you pointed out. Thus, we removed the word "significantly" from L144. We also added a note about it in L152.

180 Also relate to the change in salinity

⇒ Thank you for the important suggestion. We checked the dynamics of salinity

during the sampling period. However, salinity did not show a major change compared with nutrients and has no clear seasonality. Please see the heat map in Tables S2.

181 The lower dimilarity for viruses could be an artefact of higher taxonomic resolution.

→ Thank you for the comment. We also notice the limitation in a direct comparison of the beta diversity of viruses and prokaryotes as mentioned in the alpha diversity section.

183 The mantel results of virus/ prok and env variables seem like major findings, maybe make this a main figure/ table instead of suppl.

→ Thank you for the suggestion. According to your comment, we move the table to the main table (Table 1).

212-226 Did the observed prok. and predicted host communities have a significant mantel result?

→ Thank you for the question. These are not the results of Mantel tests. According to your comment, we performed Mantel tests for prokaryotes and viruses by the host group(taxa) in all pairs, but they were not significant in this broad-scale host range. The presumably because virus-host co-occurrence can occur in more specific virus-host pairs.

240-244 Maybe this indicates that the two abundces are likely not correlated?

→ According to the previous observations mentioned here, it might not correlate at least in some virus-host systems. However, the previous observation only discussed the reason for this but did not clarify or validate practical mechanisms. Thus, this seems to be one of the possibilities.

270 "cyanoviral" meaning cyanophages?

→ Yes. Since recently ICTV replaces "phage" with "virus" in prokaryotic virus taxon names (<https://link.springer.com/article/10.1007/s00705-015-2728-0>), we generally used the term "virus" not "phage" in the manuscript.

Figure 2 This seems like a good suppl figure, while an analysis of how well these match would be more interesting to the reader

⇒ Thank you for the suggestion but we believe it is a selling point of this study that we can assign host groups for nearly 60% of the viral community. It is generally difficult. To assign the potential host group to a majority of the viral community was an essential factor leading to the main findings of this study (prevalence of frequency-dependent infection) and we would like to retain it as the main figure.

Figure 3 I can't see correlations for Prokaryote ASVs in the figure. Also, the A, B, C, ... figure sub-labels seem to be missing. These sub-labels should be referred to in the text.

⇒ Sorry for the insufficient information on the Figure. We added the prokaryotes-prokaryotes correlations as dashed lines. Also, the sub-labels were added. Please see the revised version of Figure 3.

Figure 4 Again, this seems like a good suppl figure, while showing the correlation would be more interesting to the reader

⇒ Thank you for the suggestion. As you pointed out, Figure 3 and Figure 4 say similar host-virus correlations. However, in the analysis, we verified that the correlation is "not only the pattern of viral and host dynamics being similar but also the virus was high ranks in the viral community when the putative host is abundant". This more directly suggests host density is likely important for viral dominance. To clarify the point, we added sentences in LL250-254 but retained the figure as main figure.

298 That is why I am unsure about using the term "viral species" here, you are analysing viral contigs.

⇒ Thank you for the comments. This can influence the assessment of alpha diversity measure. We added excuses about this point in L152.

308 I would disagree, free viral particles can be fairly stable under the right conditions and drift for long times.

⇒ Thank you for the important comments. In the previous observation of the diel cycle of the viral community at the same sampling site, the viral community showed clear locality and diel cycle (Yoshida et al ISME J, 2018,

<https://doi.org/10.1038/s41396-018-0052-x>

). This likely indicates that relatively low amounts of persistent and transported particles occupy the whole community. Although free viral particles can be stable as commented, the rate of production of new viral particles seems to highly exceed the amounts of the remaining fraction at least the site.

320–324 Why would you expect a mismatch for absolute vs relative abundance correlations? Please explain a little more.

→We did not expect a mismatch between them. Absolute abundance seems to be more direct evidence, but we only have relative abundance data for most samples.

In a previous submission to a different journal, there were many comments that absolute abundance should be discussed, thus we are presenting data to support our observation that can be observed in absolute abundance level to some extent.

353–355 I can not follow this reasoning, you still don't know whether these viral contigs are infectious to the prok ASVs. eg cyanophages CAN be very host specific. Please explain. This makes the last section very speculative.

→Sorry for the incomplete explanation. We added reasonings in LL327–329 and rationale in LL347–350.

370 Probably most cellular organisms interact with multiple viruses.

→We agree with the point. However, proving it is very difficult because most of the environmental viruses are still yet host unknown. In this study, we aimed to give clues about it at least in abundant marine prokaryotic species.

390 Figure S14 is not provided. Seeing a higher diversity in more abundant viruses also seems logical/ trivial to me. Maybe your message has to be better formulated.

→Sorry, we give the wrong figure numbers for these Figures. We corrected the numbers.

Thank you for the encouraging comments. Our finding is the trend prevailed in viruses predicted to infect any host group. Please see the revised sentence in LL367–370.

482 unless I missed it, I am not clear how the rDNA sequences were converted to

cell abundance and how this was normalized among samples. For viruses the fpkm was calculated, please provide the equivalent details for cellular reads.

→ We counted prokaryotic cell numbers and virus-like particles in the several months of the samples but data for the latter half of the time series were absent because of the sample loss. Thus, to make a cut-off value for abundant OTUs, we assumed prokaryotic cell density uniformly as 10^6 cells/ml during the whole sampling period following typical coastal marine prokaryotic cell density. We added the information at LL445 to 448.

The relative abundance of each virus was calculated from the FPKM values of analyzed viruses, We rewrite L493 to make it clear.

December 29, 2022

Prof. Takashi Yoshida
Kyoto Daigaku
Graduate School of Agriculture
Sakyo-ku, Kitashirakawa-Oiwake
Kyoto, Kyoto 606-8502
Japan

Re: mSystems00931-22R1 (Prevalence of viral frequency-dependent infection in coastal marine prokaryotes revealed using monthly time series virome analysis)

Dear Prof. Takashi Yoshida:

Your manuscript has been accepted, and I am forwarding it to the ASM Journals Department for publication. For your reference, ASM Journals' address is given below. Before it can be scheduled for publication, your manuscript will be checked by the mSystems production staff to make sure that all elements meet the technical requirements for publication. They will contact you if anything needs to be revised before copyediting and production can begin. Otherwise, you will be notified when your proofs are ready to be viewed.

Publication Fees:

If you would like to submit a potential Featured Image, please email a file and a short legend to mSystems@asmusa.org. Please note that we can only consider images that (i) the authors created or own and (ii) have not been previously published. By submitting, you agree that the image can be used under the same terms as the published article. File requirements: square dimensions (4" x 4"), 300 dpi resolution, RGB colorspace, TIF file format.

We recognize that the video files can become quite large, and so to avoid quality loss ASM suggests sending the video file via <https://www.wetransfer.com/>. When you have a final version of the video and the still ready to share, please send it to mSystems staff at mSystems@asmusa.org.

Sincerely,

Michael Rappe
Editor, mSystems

Journals Department
E-mail: mSystems@asmusa.org